# OpenReview forum: "Effective and Efficient Adversarial Detection for Vision-Language Models via A Single Vector"
_ICLR.cc/2025/Conference — ICLR 2025 Conference Withdrawn Submission_

### Official Review · Reviewer_6mXm · 2024-10-29

**Soundness:** 3
**Presentation:** 3
**Contribution:** 3
**Rating:** 6
**Confidence:** 4

**Summary:**

The paper addresses the vulnerability of Visual Language Models to adversarial image attacks. The authors introduce RADAR, a dataset of adversarial images with diverse harmful responses. Additionally, they propose NEAR-SIDE, a detection method that leverages a single vector derived from VLM hidden states, known as the 'attacking direction,' to distinguish adversarial images from benign ones. Experiments on the LLaVA and MiniGPT-4 models show that the method is effective, efficient, and transferable across models.

**Strengths:**

The paper is well-structured and provides a clear motivation. The proposed method is straightforward, efficient, and effective, addressing a timely issue that underscores the need to develop robust VLM models moving forward. Additionally, the approach of extracting the 'attacking direction' (shifting from harmlessness to harmfulness) from the VLM embedding space is novel and clever. The introduction of the RADAR dataset is a valuable contribution.

**Weaknesses:**

- The method does not address adversarial attacks aimed at generating benign yet contextually altered text, which may not contain harmful content but still alters the original intent. Could the authors discuss how their approach might be extended or modified to handle such cases, where adversarial examples produce benign text that misrepresents the original context?
- The threat model needs further clarification. Could the authors define the assumed threat model more explicitly, specifying the attacker’s level of access, capabilities, and the defender's available resources? Including this in a dedicated section would enhance clarity, particularly around the assumed white-box access to the victim model.
- The method currently relies on having both adversarial and benign samples to calculate the direction and threshold. Could the authors discuss how their approach might be adapted for scenarios where only single images are available for evaluation, or clarify any limitations in these cases?
- Why does the method focus on the last layer of the LLM? Could the authors justify this choice, ideally through an ablation study comparing performance across different layers?
- While the method appears efficient based on empirical results, some visualization of the "attacking direction" would be helpful, particularly in the context of cross-model transferability.
- Only one baseline is provided for comparison, which limits the evaluation of the method's effectiveness. Could the authors explain why other relevant baselines, such as the approach in Xu et al. [1] that leverages the semantic relationship between malicious queries and adversarial images, were not included? Expanding the baseline comparisons would strengthen the evaluation in a revised version of the paper.
[1] Xu, Yue, et al. 'Defending jailbreak attack in vlms via cross-modality information detector.'"
- What if the training and testing images come from different datasets? Could the authors evaluate the robustness of their method across diverse image distributions by conducting additional experiments using separate datasets for training and testing? This would help assess the method's generalizability.
- **Line 478:** Summarize the results rather than simply directing readers to the appendix.

**Questions:**

Check Weaknesses.

---

### Official Review · Reviewer_GRGh · 2024-10-31

**Soundness:** 3
**Presentation:** 3
**Contribution:** 3
**Rating:** 3
**Confidence:** 3

**Summary:**

This paper makes two main contributions to address the under-explored problem of adversarial image attacks on Visual Language Models (VLMs). First, it introduces RADAR, a novel large-scale dataset of adversarial images that elicit diverse harmful responses from VLMs. Second, it proposes NEARSIDE, an efficient detection method that leverages a distilled "attacking direction" vector from VLM hidden states to identify adversarial images. The effectiveness of NEARSIDE is validated through extensive experiments on LLaVA and MiniGPT-4, demonstrating strong performance and cross-model transferability. While the authors acknowledge limitations in covering all possible harmful contents and the specific scope of their detection method, their work provides valuable resources and insights for improving VLM safety against adversarial image attacks.

**Strengths:**

1. The paper presents a well-structured and comprehensive approach to VLM safety.
2. The dual contribution (RADAR dataset and NEARSIDE method) provides valuable resources for the research community.
3. The methodology is clearly articulated and technically sound.

**Weaknesses:**

1. Structural Issues:
   - The organization of supplementary materials could be improved by merging Appendix A and D to enhance readability and logical flow.

2. Limited Evaluation Scope:
   - The NEARSIDE method's evaluation is confined to the proposed RADAR dataset.
   - Cross-dataset validation would strengthen the method's generalizability claims.

3. Technical Oversight:
   - The method's behavior under mixed adversarial-benign inputs (adversarial image + benign text) requires clarification.
   - Comparison with multi-modal detection methods like JailGuard needs more detailed discussion.

4. Minor Issues:
   - Typographical error in Figure 6 caption ("exemplar")

**Questions:**

1. How does NEARSIDE perform on existing adversarial image datasets beyond RADAR?
2. What is the method's response when processing combinations of adversarial images with benign prompts?
3. Could the authors elaborate on NEARSIDE's limitations compared to multi-modal detection approaches?

---

### Official Review · Reviewer_djPR · 2024-11-02

**Soundness:** 2
**Presentation:** 2
**Contribution:** 3
**Rating:** 6
**Confidence:** 4

**Summary:**

This paper introduces an innovative and efficient approach for detecting adversarial images in Vision-Language Models (VLMs), addressing their vulnerability to adversarial attacks. Current datasets and detection methods face limitations, either lacking diversity or being computationally heavy, particularly with low-visibility attacks. Key contributions of this work include a novel method for identifying the attacking direction in VLMs’ hidden space, which serves as a defense mechanism against adversarial images. This approach is integrated into the NEARSIDE method, which is efficient, requiring only a single forward pass and showing cross-model transferability. The authors also constructed the RADAR dataset, comprising 4,000 high-quality adversarial samples, surpassing previous datasets in scale and harmful response diversity. Through experiments on LLaVA and MiniGPT-4, NEARSIDE demonstrates high accuracy and significant speed improvements over baseline methods, highlighting its potential to enhance VLM security. While promising, further investigation could strengthen its generalizability across different VLMs.

**Strengths:**

This research presents an innovative approach to detecting adversarial images by pinpointing the attacking direction within the hidden space of Vision-Language Models (VLMs). By focusing on this attacking direction, the method offers a fresh perspective that contrasts with traditional techniques, which often emphasize response discrepancies or image purification. Additionally, the introduction of the RADAR dataset, which captures a wide range of harmful responses, fills a crucial gap in evaluating VLM safety and significantly enhances the quality of related research.

The NEARSIDE method is thoughtfully designed to effectively distinguish between adversarial and benign inputs, demonstrating impressive accuracy and making it practical for real-time applications. Rigorous experiments, including comparisons with the leading JailGuard baseline on the RADAR dataset, confirm the method's effectiveness. The exploration of cross-model transferability and varying perturbation radii further highlights a deep understanding of how the method performs under different conditions.

Clarity is a strong point of this paper, as it articulately explains the vulnerabilities of VLMs and the urgent need for effective detection strategies. The description of the NEARSIDE method is straightforward, supported by clear illustrations that guide readers through the process. Furthermore, the well-structured presentation of the experimental setup and results makes it easy to follow, allowing readers to appreciate the significance and implications of the research findings.

**Weaknesses:**

1.The concept of attacking direction is intriguing, yet the paper does not adequately address its stability across varying training datasets and model architectures. This is particularly relevant given that visual language models (VLMs) are frequently updated in practice. I suggest conducting experiments to evaluate how changes in training data or slight architectural modifications impact the stability of the attacking direction. A deeper analysis of these factors could enhance the understanding of NEARSIDE's detection performance.

2.The evaluation of the proposed method is primarily limited to LLaVA and MiniGPT-4, raising concerns about its applicability to other VLMs and LLMs with different architectures. While the initial cross-model transferability analysis is a good starting point, a more extensive evaluation is necessary. I recommend testing NEARSIDE across a broader spectrum of models, particularly those with varied visual encoders and training methodologies, to better assess its generality and potential need for modifications.

3.The current focus on detecting adversarial images generated by existing techniques overlooks the possibility of attackers developing new strategies to evade detection. As the landscape of adversarial attacks evolves, it is crucial to evaluate NEARSIDE's resilience against these future threats. Conducting simulations of more sophisticated adaptive attacks could provide insights into potential countermeasures and enhance the robustness of the detection method.

4.The paper acknowledges that the detection threshold, determined from the training set, may not be optimal across all datasets. However, it lacks a thorough investigation into dynamic threshold adjustment methods. I recommend exploring adaptive thresholding techniques or statistical analyses that could lead to more reliable threshold settings. Additionally, a detailed examination of how different thresholds affect false positive and false negative rates would be beneficial.

5.Although NEARSIDE is claimed to be efficient relative to baseline methods, the computational complexity analysis is somewhat superficial. The current focus on inference time neglects other critical factors, such as memory usage and training costs associated with extracting the attacking direction. A more comprehensive breakdown of computational costs throughout the process is essential. Discussing potential optimizations could further enhance NEARSIDE's scalability and performance in practical applications.

**Questions:**

No further questions; the suggestions have been fully covered in the **Weaknesses** section.

---

### Official Review · Reviewer_TrC1 · 2024-11-04

**Soundness:** 1
**Presentation:** 2
**Contribution:** 1
**Rating:** 1
**Confidence:** 4

**Summary:**

This paper proposes a new method to detect jailbreaking attacks against visual language models (VLMs). The method is based on the observation that large language models contain a set of steering vectors in the intermediate embeddings that can be modulated to generate texts towards certain specific attributes. Based on this observation, the authors propose to (1) build a larger dataset of adversarial attacks, and (2) use the dataset to learn an attack direction in steering vectors as indication of attacks. To validate the idea, the authors created a dataset of 4000 samples, include a 500 sample training set and three test sets. Training set uses images from COCO validation set and queries from the train set of HH-rlhf harm-set. Test sets use COCO test set and queries from the test set of HH-rlhf harm-set, D-corpus, and Harm-Data.

**Strengths:**

+ The proposed method is based on an observation cross-validated from multiple previous studies
+ The proposed method outperformed a baseline

**Weaknesses:**

The evaluation design is very flawed.
- The evaluation does not include performance on benign queries. As a result, how likely the method generate false alarms is unknown.
- While the RADAR dataset is larger than previous ones and is generated using different attack queries, all the images are generated in the same way. As a result, the experiment results may not generalize to other attack methods.
- The evaluation also lacks experiments on adaptive attacks, i.e., whether it's possible to generate adversarial images that would lead to harmful output yet does not go beyond the learned threshold.

**Questions:**

* What is the false detection rate on common benign datasets (e.g., VQA)?
* What is the detection rate against other attack methods?
* Can the proposed method detect adaptive attacks?

---

### Note · Authors · 2024-11-14

I have read and agree with the venue's withdrawal policy on behalf of myself and my co-authors.